# Surface Enhanced Raman Spectroscopy Detection of Sodium Thiocyanate in Milk Based on the Aggregation of Ag Nanoparticles

**DOI:** 10.3390/s19061363

**Published:** 2019-03-19

**Authors:** Yanting Feng, Rijian Mo, Ling Wang, Chunxia Zhou, Pengzhi Hong, Chengyong Li

**Affiliations:** 1College of Food Science and Technology, Guangdong Ocean University, Zhanjiang 524088, China; yanting_feng@163.com (Y.F.); rijian_mo@163.com (R.M.); chunxia.zhou@163.com (C.Z.); hongpengzhi@126.com (P.H.); 2Shenzhen Institute of Guangdong Ocean University, Shenzhen 518108, China; 3School of Chemistry and Environment, Guangdong Ocean University, Zhanjiang 524088, China; 4Coastal Ecology Engineering Technology Research Center of Zhanjiang City, Guangdong Ocean University, Zhanjiang 524088, China

**Keywords:** surface-enhanced Raman scattering (SERS), milk, Sodium thiocyanate (NaSCN)

## Abstract

A method is developed for detecting the concentration of sodium thiocyanate (NaSCN) in milk based on surface-enhanced Raman scattering (SERS) technology. A trichloroacetic acid solution can be used to enhance the SERS signal because of its function in promoting the aggregation of Ag nanoparticles (Ag NPs). Meanwhile, the protein in milk would be precipitated as trichloroacetic acid added and the interference from protein could be reduced during the detection. In this work, the enhancement factor (EF) is 7. 56 × 10^5^ for sodium thiocyanate in water and the limit of detection (LOD) is 0.002 mg/L. Meanwhile, this method can be used to detect the concentration of sodium thiocyanate in milk. Results show that SERS intensity increased as the concentration of sodium thiocyanate increase from 10 to 100 mg/L. The linear correlation coefficient is R^2^ = 0.998 and the detection limit is 0.04 mg/L. It is observed that the concentration of sodium thiocyanate does not exceed the standard in the three kinds of milk. The confirmed credibility of SERS detection is compared with conventional methods.

## 1. Introduction

Sodium thiocyanate (NaSCN) is a chemical used in medicine, printing and dyeing, chemistry, etc. [1,2,3,4,5]. To activate milk’s advanced antibacterial system, the addition of trace amounts of NaSCN is the key [6]. NaSCN has a good natural antibacterial effect and low price, so the excessive addition of sodium thiocyanate still exists [7]. If excessive amounts of sodium thiocyanate are added to the milk, it will be harmful to the human body [6,7,8,9,10]. Therefore, the detection of NaSCN is of great importance.

To detect thiocyanate in milk, various methods have been reported including spectrophotometric, high-performance liquid chromatography (HPLC), etc. [11,12,13,14,15,16]. Although these methods have the advantages of high accuracy and sensitivity, they require expensive instruments and professional technicians, and they are subjected to complicated pretreatment long time consumption. Thus, it is urgent to develop a new method of rapidly detection NaSCN in milk.

Surface-enhanced Raman scattering (SERS) is one of the most popular tools for biosensor and chemical analysis [17,18,19,20,21]. It enables ultrasensitive detection technology to prove the ability to identify trace molecules and can be applied to the detection of toxins in food [22,23,24,25,26,27]. The matrix plays a vital role in SERS detection. Wu et al. [28] used a core-satellite substrate to detect melamine and NaSCN in milk and the limit of detection was 5.8 μg/L. The SERS matrix was fabricated via immobilized Au nanoparticles (AuNPs) on polymer beads to form core-satellite nanostructures and the process was complicated. Zhang et al. [12] developed a method to detect NaSCN content in milk based on agarose SERS microchips and the limit of detection was 0.5 mg/L. However, the microchip was needed and expensive equipment was required in the experiment. Ag NPs is a commonly used SERS active substrate. Some research has shown that the addition of substances such as salts and acids into Ag NPs would lead to the aggregation of nanoparticles, thereby generating a rich SERS hot spot and enhancing the Raman signal [13]. 

In this work, a method is developed for detecting the concentration of NaSCN in milk based on SERS method. Trichloroacetic acid is added to promote the precipitation of protein and the aggregation of Ag NPs. As a consequence, the Raman signal is enhanced and the interference from protein is reduced. The proposed method has been used in the detection of sodium thiocyanate for three kinds of milk in the supermarket.

## 2. Materials and Methods

### 2.1. Reagents and Materials

Silver nitrate (AgNO_3_) was bought from Beijing Huawei Ruike Chemical Co., Ltd. (Beijing, China). Sodium citrate (Na_3_C_6_H_5_O_7_·2H_2_O), sodium thiocyanate (NaSCN) and trichloroacetic acid (C_2_HCl_3_O_2_, TCA) were supplied by Shandong West Asia Chemical Industry Co., Ltd. (Shandong, China). Hydrochloric acid (HCl) and nitric acid (HNO_3_) were purchased from Lianjiang Ai Lianhua Reagent Co., Ltd. (Guangdong, China). All other reagents were analytically pure and used without purification. 

Electronic analytical balance (A JJ124BC) purchased from Shenzhen Langpu Electronic Technology Co., Ltd. (Shenzhen, China). Aggregating heat constant temperature blender with magnetic force purchased (DF-101S) from Gongyi Honghua Instrument Equipment Industry & Trade Co., Ltd. (Zhengzhou, China). UV-Visible absorption spectra (Hitachi, U-3900H) were obtained with a spectrometer. (Beijing, China). High-resolution transmission electron microscopy (JEOL, JEM-1400, Akishima City, Tokyo, Japan) was performed at 120 kV to determine the sizes of Ag NPs. Spectral signals were acquired using a portable Raman spectrometer (Ocean Optics, SR-510 Pro, Beijing, China).

### 2.2. Preparation of Ag NPs

A total of 100 mL of 0.18 g/L silver nitrate was added to a clean round bottom flask. After the liquid heated to boiled, 2 mL of 1% sodium citrate solution was quickly added and reacted for 1 h. When the solution eventually turned grayish yellow, the heating was stopped. After the silver colloid was cooled down, it was stored at 4 °C in the dark. 

### 2.3. Observation of Ag NPs

To initially determine the particle size of Ag NPs, Ag NPs were diluted 10 times with ultrapure water, and then the maximum ultraviolet absorption wavelength of Ag NPs was measured using a UV-Visible spectrophotometer at a wavelength of 300–700 nm.

Before characterization by transmission electron microscopy, 9 mL of 15% TCA was added to 1 mL of Ag NPs to generate aggregation of Ag NPs.

### 2.4. Preparation of the Samples

Preparation of NaSCN standard solution in milk: 30 μL of different concentrations (0, 10, 30, 50, 80, 100 mg/L) of NaSCN standard solution was added to 270 μL of milk, then 900 μL of 15% trichloroacetic acid was added. After that, the mixture was subjected to centrifugal treatment for 10 min (4 °C, 12,000 rpm/min), with 120 μL of supernatant and 10 μL of Ag NPs mixed in the centrifuge tube. 

The detection of NaSCN in milk samples: 300 μL of three kinds of milk was added into a 1.5 mL centrifuge tube containing 900 μL of 15% trichloroacetic acid, respectively. Then samples were mixed thoroughly (for 1 min), followed by centrifugation for 10 min (4 °C, 12,000 r/min). A total of 120 μL of supernatant and 10 μL of Ag NPs were thoroughly mixed and let stand for 6 min before Raman detection.

### 2.5. Detection of NaSCN with SERS

In SERS measurement, the wavelength of excitation laser was 785 nm and the power was set to 350 mW. The spectrum was collected from 170 to 3900 cm^−1^ with a spectral resolution of 4 cm^−1^. The probe working distance was 7.5 mm and the spot diameter was less than 2 mm. In this experiment, SERS was measured at 30% power (105 mW) and a single 15 s accumulation. It was detected in the dark to minimize the signal from the fluorescent lights. Data acquisition and analysis were performed using the spectrometer’s AccuRam software.

### 2.6. Detection of NaSCN with UV-Visible Absorption

A total of 15 mL milk and 10 mL TCA (200 g/L) were added into a centrifuge tube and shocked vigorously. It was settled for 30 min. After centrifugation (9000 rad/min, 3min), the supernatant was filtered. A total of 4 mL filtered solution was added to a 10 mL colorimetric tube, followed by addition of 2 mL ferric nitrate solution and reacted for 10 min. Then it was measured using a UV-Visible spectrophotometer at a wavelength of 450 nm. 

## 3. Results and Discussions

### 3.1. Characterization of Ag NPs

The UV-Vis spectrum of the Ag NPs is shown in Figure 1a. The morphology, size and distribution of the Ag NPs could be characterized by the position and peak shape of the maximum absorption. There is an Ag NPs maximum ultraviolet absorption at 405.5 nm, which corresponds to the Ag NPs diameter of about 55 nm. It can be seen that the peak shape of the peak is narrow, indicating that the Ag NPs prepared by the method is mostly spherical and has a uniform particle size distribution. Most of Ag NPs present are as a monomer before the addition of coagulant (Figure 1b). The shape of Ag NPs is spherical. The particle size distribution is relatively uniform and the majority of the particle size distribution of Ag NPs is 55 nm (Figure 1c). The diameters of Ag NPs are almost identical. After the addition of the coagulant TCA, the potential balance of the colloid is destroyed, leading to the aggregation of Ag NPs. As can be observed in Figure 1d, the aggregated Ag NPs has a richer SERS hot spot, which can enhance the Raman signal of NaSCN.

### 3.2. SERS Detection of NaSCN

As shown in Figure 2a, there are two characteristic peaks of NaSCN powder at 753 cm^−1^ and 2075 cm^−1^, respectively. The absorption peak at 753 cm^−1^ is caused by the C-S bond stretching vibration. The other absorption peak at 2075 cm^−1^ is attributed to the antisymmetric stretching vibration of -C≡N bond, which is chosen as the characteristic peak for identifying of NaSCN due to its high intensity [29]. In the experiment, the peak position of SCN^−^ in the solution is shifted to 2126 cm^−1^ compared to the one in sodium thiocyanate powders (2075 cm^−1^). It has been reported that when thiocyanate ions are adsorbed on the surface of Ag NPs, the peak position will change due to the different adsorption modes of ions in solution [30]. When NaSCN is added, Ag NPs are aggregated (Figure 2b). The Raman signal is hardly found in 10^4^ mg/L NaSCN even Ag NPs were added. After the addition of TCA to the above solution, an obvious Raman signal can be observed. This indicates that the aggregation of Ag NPs caused by the coagulant TCA has a good reinforcing effect on the Raman signal of NaSCN. There is a perfect gap between the condensed Ag NPs, which provide a mass of SERS hot spots. When the NaSCN molecule is adsorbed to the gap of the nanoparticles, the Raman signal of NaSCN would be greatly enhanced. In order to investigate whether the addition of TCA interferes with the SERS signal of NaSCN, the SERS spectrum of the mixed solution of TCA and Ag NPs was verified in the experiment. As depicted in Figure 2b, there is no obvious characteristic peak at a wave number of 1800~2500 cm^−1^ in pure TCA solution or a mixed solution of TCA and Ag NPs. Thus, they do not interfere with the Raman signal of NaSCN.

When TCA is added into an Ag NPs solution, Ag NPs aggregate and SERS hot spots appear. The reaction time is important in this process. Figure 3 shows the relation curve between the SERS spectra and the reaction time. In the beginning, the hot spots appear and the intensity increases as TCA added. At about 6 min, the intensity reaches the maximum. However, this reduces as the time increases gradually. Because a few of Ag NPs precipitate in the bottom, the SERS hot spots decrease slightly when the reaction time is too long.

### 3.3. SERS Detection of NaSCN in Aqueous Solution

As shown in Figure 4a, a distinct characteristic peak at 2126 cm^−1^ in 0.1 mg/L NaSCN solution is indicated. The intensity of Raman signal (2126 cm^−1^) rises with the increase in the NaSCN concentration. There has a linear range from 0.1 mg/L to 1 mg/L and LOD is 0.002 mg/L (Figure 4b).

In order to calculate the enhancement factor of the Ag NPs, a conventional Raman signal with a concentration of 10^4^ mg/L aqueous NaSCN and a SERS Raman signal in a concentration of 1 mg/L were experimentally tested. The intensity of conventional Raman signal at 2126 cm^−1^ is 156 counts in 10^4^ mg/L NaSCN solution, while it can approach 11,794 counts in 1 mg/L NaSCN solution. The following is the calculation formula for the enhancement factor (EF) of Ag NPs in aqueous solution:
$$\mathrm{EF}=\frac{I_{SERS}}{I_{NR}}\times \frac{C_{NR}}{C_{SERS}}$$
where *I_SERS_* is the characteristic peak intensity of the surface-enhanced Raman spectrum; *I_NR_* is the characteristic peak intensity of the conventional Raman spectrum; *C_SERS_* is the sample concentration of the surface-enhanced Raman spectrum; *C_NR_* is the sample concentration of the conventional Raman spectrum. Using the formula, the enhancement factor of Ag NPs is calculated to be 7.56 × 10^5^.

### 3.4. SERS Detection of NaSCN in Spiked Milk Samples

In milk examples, there are mass of proteins, which hinder the detection of sodium thiocyanate. In this experiment, TCA is used to promote protein precipitation and gather Ag NPs simultaneously, without the requirement of additional processing during the measurement. As shown in Figure 5a, the concentrations of NaSCN were measured by SERS in the milk. The proteins contained in the milk would adsorb on the surface of the Ag NPs, which hinder the contact between thiocyanate and Ag NPs. Meanwhile, a portion of thiocyanate is absorbed by protein and thus precipitates upon the addition of TCA. 

As seen in Figure 5b, the Raman signal of different concentrations (0, 10, 30, 50, 80, 100 mg/L) of thiocyanate in milk was tested. With the range of 10–100 mg/L, the Raman intensity of NaSCN follows the linear equation *I_SERS_* = 74.133*C_NaSCN_* − 65.8 with the linear correlation coefficient R^2^ = 0.998. The calculated LOD value is 0.04 mg/L (LOD = 3SD/k, where SD and k are the standard deviation of the blank and the slope of the calibration graph respectively).

As showed in Figure 6, we tested three kinds of milk samples that are commonly found in the market, named sample A, sample B, sample C. There is no obvious peak at 2126 cm^−1^ for the three kinds of milk samples compared to that of the NaSCN spiked milk. Thus, it can be known that the concentration of NaSCN in the milk is less than 10 mg/L.

### 3.5. Determination of NaSCN in Milk by UV-Visible Absorption Spectrometry

In order to verify the accuracy of SERS detection, UV-visible absorption spectrometry was employed to measure the concentration of NaSCN in the milk. Figure 7 shows the standard curve measured by the UV-visible absorption spectrometry. There is an excellent linear relationship between the absorption and the concentration of NaSCN (R^2^ = 0.999), with a LOD of 0.069 mg/L. As shown in Table 1, the absorption is very weak at 450 nm for three milk samples. The concentration of NaSCN is 0.172, 0.092, 0.813 mg/L for sample A, B and C, respectively. Thus, they are much lower than 10 mg/L, which is accordance with the results of SERS detection. 

## 4. Conclusions

In this work, a method is developed for detecting the concentration of NaSCN in milk based on SERS method. Trichloroacetic acid is used to promote protein precipitation and gather Ag NPs simultaneously, without the requirement of additional processing during the measurement. The detection limit of NaSCN in water and milk is 0.002 mg/L and 0.04 mg/L, respectively. The proposed method achieves good linearity (R^2^ = 0.998) within the range of 10 to 100 mg/L of sodium thiocyanate in the spiked milk. In our verification test, the LOD of SERS (0.04 mg/L) was lower than the UV-visible absorption spectrometry (0.069 mg/L) which confirms the credibility of SERS when used for detection of NaSCN in milk. Therefore, it can be used to measure the concentration of sodium thiocyanate in milk quickly.

## Figures and Tables

**Figure 1 sensors-19-01363-f001:**
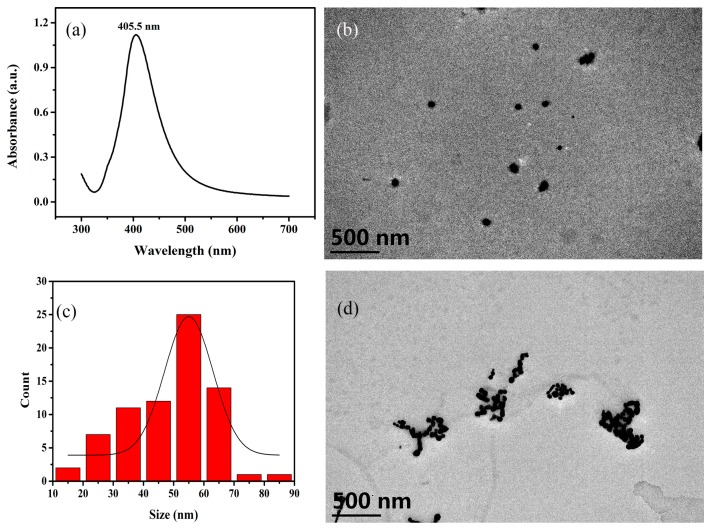
The characteristics of Ag NPs. (**a**) UV-Vis spectrum of Ag nanoparticles (NPs) as prepared. (**b**) Transmission electron microscope (TEM) image of Ag NPs as prepared. (**c**) Particle size distribution of Ag NPs (**d**) TEM image of Ag NPs after the addition of trichloroacetic acid (TCA).

**Figure 2 sensors-19-01363-f002:**
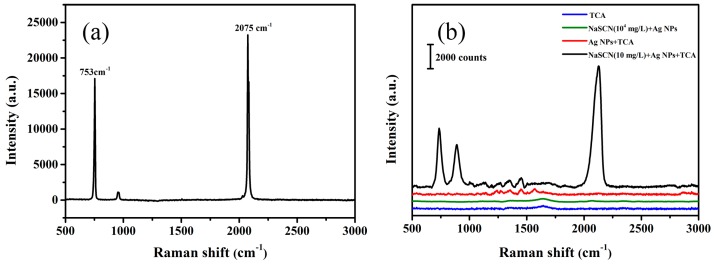
The Raman spectra of the samples. (**a**) NaSCN solid powders. (**b**) TCA, a mixed solution of TCA and Ag NPs, a mixed solution of NaSCN and Ag NPs, a mixed solution of NaSCN, TCA and Ag NPs.

**Figure 3 sensors-19-01363-f003:**
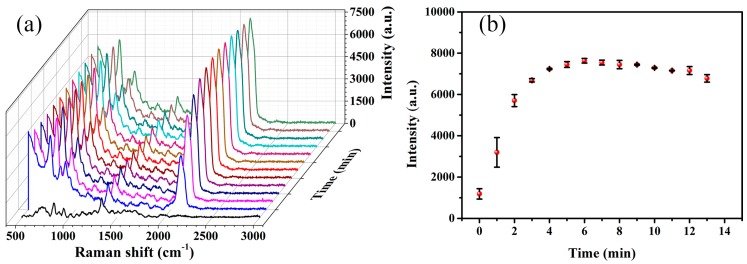
The relationship between surface-enhanced Raman scattering (SERS) spectra and the reaction time (repeated three times). (**a**) SERS spectra with different reaction times. (**b**) The relation curve between Raman spectra intensity (2126 cm^−1^) and the reaction time.

**Figure 4 sensors-19-01363-f004:**
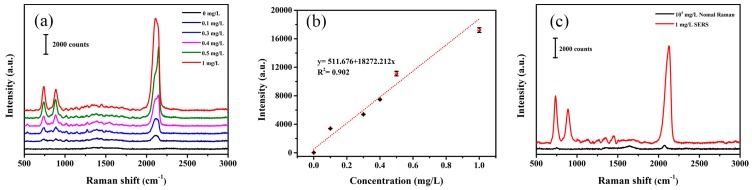
The SERS Detection of NaSCN in an aqueous solution (repeated three times). (**a**) The SERS spectra of NaSCN aqueous solution with different concentrations. (**b**) The relation curve between the intensity of SERS at 2126 cm^−1^ and the concentration of NaSCN. (**c**) The SERS spectrum of 1 mg/L NaSCN aqueous solution and conventional Raman spectrum of 10^4^ mg/L NaSCN aqueous solution.

**Figure 5 sensors-19-01363-f005:**
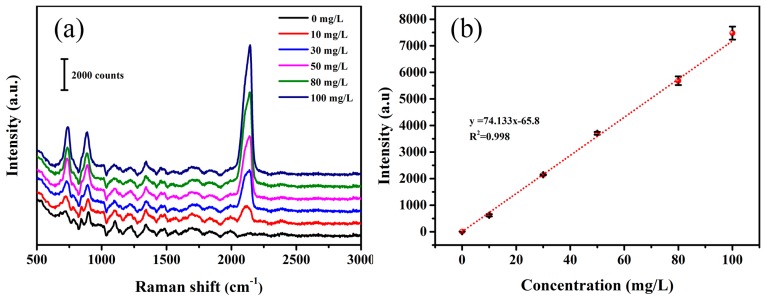
The SERS detection of NaSCN in the milk (repeated three times). (**a**) SERS spectra of the milk mixed with different concentrations of NaSCN. (**b**) The relation curve between the intensity of SERS at 2126 cm^−1^ and the concentration of NaSCN.

**Figure 6 sensors-19-01363-f006:**
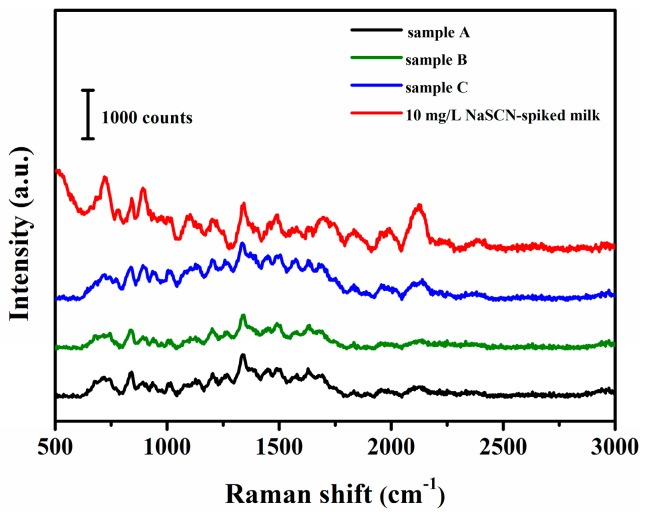
The SERS spectra of three kinds of commercial milk samples and spiked milk.

**Figure 7 sensors-19-01363-f007:**
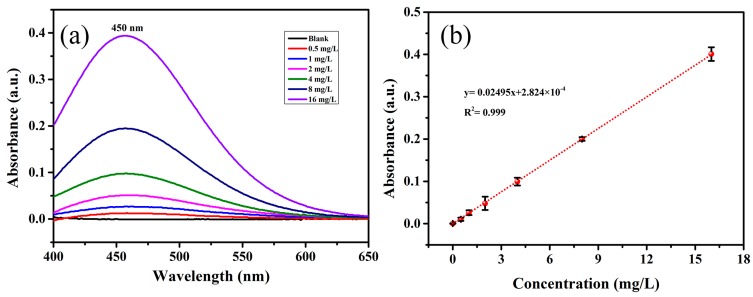
The detection of NaSCN in the milk via UV-visible absorption spectrometry (repeated three times). (**a**) UV-visible absorption of NaSCN with different concentrations in the milk. (**b**) The relation curve between the absorption at 450 nm and the concentration of NaSCN.

**Table 1 sensors-19-01363-t001:** The concentrations of NaSCN in three milk samples (UV-visible absorption spectrometry vs. surface-enhanced Raman scattering (SERS)) (repeated for three times).

	Methods	UV-Visible Absorption Spectrometry Absorbance (λ = 450 nm)	UV-Visible Absorption Spectrometry (mg/L)	SERS Detection (mg/L)
Samples				
Sample A		0.011 ± 0.001	<10	<10
Sample B		0.006 ± 0.001	<10	<10
Sample C		0.051 ± 0.002	<10	<10

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
