# Peer review of "Surface Enhanced Raman Spectroscopy Detection of Sodium Thiocyanate in Milk Based on the Aggregation of Ag Nanoparticles"

_sensors, 2019, doi:10.3390/s19061363_

Reviewer 1 Report

Feng et al. present interesting work that details a method for the aggregation of silver nanoparticles in order to detect of toxic contaminants in milk through surface-enhanced Raman scattering (SERS) spectroscopy. They provide impressive results for the detection of sodium thiocyanate, including a highly linear calibration curve for its detection in milk. The work is potentially applicable as a simple, low-cost method for rapid analysis of milk products. However, the work is fairly incomplete; the results that are proposed in the abstract do not appear in the manuscript. Moreover, the writing style of the manuscript makes it very challenging to follow as a reader. The work would benefit from the following edits.

 Major Comments:

 General

·         This work would benefit from thorough proofreading. There are many errors that make the manuscript challenging to read.

·         The manuscript proposes results of melamine detection (see the abstract and the title of Section 3.2). However, these results do not appear anywhere in the manuscript.

·         The authors use the term “interferon” regularly instead of “interference” when referring to spectral interference from the proteins in milk.

Abstract

·         The abstract discusses a limit of detection of NaSCN in both water and milk, but only the milk calibration curve is shown in the manuscript. If this is a key finding validating inclusion in the abstract it should certainly appear in the manuscript.

 Introduction

·         The first two sentences should be rewritten for clarity. “Chemistry raw” should be replaced by “chemical”.

 Experimental

·         The authors should state what type of Raman spectrometer was used.

 Results

·         The authors should provide citations for peak assignments in the identified Raman spectra.

Author Response

It is in the attachment.

Reviewer 2 Report

Dear Editor,

I would recommend its publication after the following major revisions:

1)     The Authors do not specify whether it is a new method or modification of an existing one. In both cases, a better literature review should be made in order to compare the results obtained with other measurement methods.

2)     Checking the measurement method with the NaSCN measurement in milk purchased in the market has not been carried out properly. The NaSCN level in market milk should be confirmed by other measurement methods (e.g. HPLC) so that the obtained result could be considered appropriate.

3)     The manuscript lacks information on possible factors affecting the accuracy of the measurement. Determining the sensitivity of the method with only an aqueous solution is not entirely reliable. The result of the measurement indicating that there is very low NaSCN concentration in the market milk (less than 10 mg / L) may not be adequate. An in-depth discussion of the result obtained is essential.

Author Response

It is in the attachment.

Reviewer 3 Report

Comments to the author:

In this manuscript the authors proposed a new methodology for determining sodium thiocyanate (NaSCN) in milk samples. The methodology is based on the SERS detection of NaSCN by using a controlled aggregation of Ag nanoparticles mediated by trichloroacetic acid. They performed the synthesis and characterization of Ag nanoparticles as well as the detection limit of NaSCN first by using aqueous NaSCN and then with a matrix of milk samples. They also tested the NaSCN content of three different milks available in the market. Overall, this contribution presents a facile, and sensitive fabrication of a SERS methodology which shows great potential in the detection of sodium thiocyanate (NaSCN) in milk samples. I would recommend the publication of this contribution but some issues should be addressed.

The following are some questions and suggestions for improving their work:

Introduction

1.       In line 45 you mention two works that detect NaSCN but you did not mention neither the LOD of NaSCN nor the advantages of your methodology compared to this works.

Methodology

1.       After the addition of trichloroacetic acid to the milk samples how much time do you wait before the centrifugation step?

2.       Later, when you describe the methodology for detection NaSCN in real milk samples you did not mention the addition of trichloroacetic acid.

Results

1.       Size of nanoparticles should contain a standard deviation

2.       What I miss most here is the kinetics of aggregation of the Ag nanoparticles mediated by trichloroacetic acid. You said that you wait 6 minutes before measuring the SERS spectra but how are the kinetics of aggregation and the reproducibility?

3.       The shown results for the LOD are from how many measurements and from how may experiments? Either with aqueous solutions or spiked in milk samples I can not see the standard deviations. Reproducibility should be proved.

4.       How do you calculate the LOD of NaSCN? Could you indicate that?

Minor issues

1.       Line 111—Error it is said SERS detection of Melamine

2.       Line 135—Error it is said Raman detection instead of SERS detection. Here you said Raman spectra and it is confusing because figure 3a is SERS and in figure 3b you are comparing with Raman detection. The same for the caption of the figure. And also for line 156 and caption figure 4.

Author Response

It is in the attachment.

Round  2

Reviewer 1 Report

Feng et al. present interesting work that details a method for the aggregation of silver nanoparticles in order to detect NaSCN in milk through surface-enhanced Raman scattering (SERS) spectroscopy. They provide impressive results for the detection of sodium thiocyanate, including a highly linear calibration curve for its detection in milk. Despite some revisions, the writing style of the manuscript still makes it challenging to follow as a reader. The work would benefit from the following edits. I recommend acceptance of the manuscript with the following major and minor revisions.

Major Comments:

Results

·       The authors should analyze the data from the UV-vis detection experiment to obtain an LOD and compare this to the LOD from the SERS experiment. The conclusion made in line 205: “Thus, the concentrations of NaSCN are lower than 10 mg/L…” is not an appropriate statement to make until you have determined the LOD of the UV-vis system.

Minor Comments:

 General

·       This work would still benefit from thorough external proofreading. There are many errors that make the manuscript challenging to read.

·       Repeating comment 3 from the previous review round, the authors use the term “interferon” regularly instead of “interference” when referring to spectral interference from the proteins in milk. See line 20 and line 54.

 Abstract

·       Line 22: “Meanwhile, this method can be used to detect the concentration of melamine in milk”. The authors do not apply this method for the detection of melamine in milk. This statement should be removed from the abstract.

 Introduction

·       Line 47: “the limit of detection was as low as…” should be reworded to explicitly state the limit of detection.

Author Response

It is in the attachment.

Reviewer 2 Report

Manuscript is ready for publication

Author Response

English language and style  were corrected in the revised manuscript.